# Transcriptional Profiling Reveals Adaptive Response and Tolerance to Lactic Acid Stress in *Pichia kudriavzevii*

**DOI:** 10.3390/foods11182725

**Published:** 2022-09-06

**Authors:** Hai Du, Yan Fu, Nan Deng, Yan Xu

**Affiliations:** Laboratory of Brewing Microbiology and Applied Enzymology, Key Laboratory of Industrial Biotechnology of Ministry of Education, School of Biotechnology, Jiangnan University, 1800 Lihu Avenue, Wuxi 214122, China

**Keywords:** *Pichia kudriavzevii*, mRNA sequencing, lactic acid stress, response pathway, type strain

## Abstract

*Pichia kudriavzevii* plays an important role in fermented foods and beverages. In the long domestication process of traditional fermentation, the mechanism of response to lactic acid, a common metabolite and growth inhibitor, is currently unclear in *P. kudriavzevii*. In this study, the tolerance to lactic acid of *P. kudriavzevii* C-16, isolated from fermented grains, was compared with its type strain ATCC 24210. Under lactic acid stress, *P. kudriavzevii* C-16 showed increased biomass yields and lactic acid consumption rates. Then, mRNA sequencing was used to analyze the response to lactic acid in *P. kudriavzevii* C-16. Results showed that 92 and 96 genes were significantly upregulated, 52 and 58 genes were significantly downregulated, respectively, in *P. kudriavzevii* C-16 cultured for 12 h and 24 h. The genes, which involved in pyruvate metabolic pathway, ABC transporter proteins, glutamate metabolic pathway, and the biosynthetic pathway of leucine and valine, were observed to be differentially expressed between the *P. kudriavzevii* C-16 and its type strain ATCC 24210. By analyzing the production of higher alcohols, the concentrations of isobutyl alcohol and isoamyl alcohol produced by *P. kudriavzevii* C-16 increased significantly. It was consistent with the up-regulation of genes that biosynthesized related amino acids.

## 1. Introduction

In traditional food and beverage fermentation, long-term specific fermentation environment leaded to adaptive evolution of microorganisms [1,2]. During domestication, microorganisms acquired the ability to efficiently consume certain nutrients, respond to numerous industry-specific stressors, and produce desirable compounds [3]. This has significant advantages in process control in terms of product repeatability, safety, taste and flavour. It is also of great significance to the breeding of fine microorganism and the enzymatic modification of specific genes.

Chinese soy sauce-flavour *Baijiu* production is a spontaneous, simultaneous saccharification and fermentation process [4]. It involves various microorganisms and complex metabolites [4,5]. Among them, lactic acid (up to 36.20 ± 6.20 g/kg fermented grain) and acetic acid (up to 23.98 ± 7.18 g/kg fermented grain) are major acids [6]. This high-acid fermentation environment can inhibit the growth of natural microorganisms and is bad for their proliferation. Therefore, we hypothesized that the core functional strains would undergo adaptive domestication in this environment to ensure their survival and normal fermentation. This hypothesis has been tested in other fermented foods. After propagation of wild *Lactococcus lactis* strains originated from plant niche for 1000 generations in milk, evolved strains displayed significantly increased acidification rates and biomass yields in milk, and lost a transposon containing genes important in the plant niche but dispensable in milk [7]. A more clear example of a domesticated fungus is *Aspergillus oryzae*, which is important for the production of traditional fermented foods and beverages in Japan [8]. It has been postulated that *A. oryzae* was domesticated from an *A. flavus* variant that no longer produced the carcinogenic toxin aflatoxin [8]. In *A. oryzae*, only two expressed sequence tags (ESTs) of the aflatoxin pathway genes were detected, whereas they were all found in *A. flavus* [8]. However, the most intensively studied microbial domesticate is probably the baker’s and brewer’s yeast *Saccharomyces cerevisiae* [3]. Therefore, studying the molecular mechanism of tolerance of the core functional strains such as *Saccharomyces cerevisiae* and *Pichia kudriavzevii* under the conditions of interest can better understand the role of them in the fermentation system. It can provide some theoretical basis for the more reasonable application of them in fermentation systems of fermented foods and beverages.

In this study, we compared the capability of lactic acid tolerance of *P. kudriavzevii* C-16, isolated from fermented grains, with the type strain ATCC 24210, and explored the response mechanisms to the high lactic acid stress in *P. kudriavzevii* C-16. Based on the whole genome sequence of *P. kudriavzevii* CBS573, with the aid of RNA-Seq technology, using the type strain *P. kudriavzevii* ATCC 24210 as a control, the gene transcription affected by lactic acid in *P. kudriavzevii* C-16 was revealed. It is expected to guide the effective use of *P. kudriavzevii* in food and beverage fermentation.

## 2. Materials and Methods

### 2.1. Strains, Media and Culture Conditions

*P. kudriavzevii* C-16 was isolated from the soy sauce-flavour *Baijiu* fermented grains and deposited in the China General Microbiological Culture Collection Center as CGMCC 19337. The type strain *P. kudriavzevii* ATCC 24210 (=CGMCC 2.1465) was provided by the China General Microbiological Culture Collection Center. They are referred to as the *P. kudriavzevii* C-16 and the type strain ATCC 24210 respectively in this article. All the yeasts, stored at −80 °C in yeast extract-peptone-dextrose (YPD) glycerol stock, were incubated on YPD agar plates at 30 °C for 24 h to restore the activity of the strains for subsequent seed culture.

The seed and fermentation medium were both sorghum extract medium. We prepared the sorghum extract medium as follows. 400 g broken sorghum was soaked in 1.6 L water overnight. Next, we added appropriate amount of thermostable α-amylase (4 × 10^4^ U/g) into the mixture. The mixture was sterilized at 105 °C for 1 h. After that, added the α-amylase again and cooked for 3 h. Subsequently added appropriate amount of glucoamylase (1 × 10^5^ U/g) when the temperature dropped to 60 °C. We saccharified the mixture at 60 °C for 4 h. At last, we used two layers of gauze to filter the amylolytic mixture and centrifuged it at 4000× *g* for 10 min to gain the supernate. The sorghum extract was the collected supernate. We dilute sugar concentration of the obtained sorghum extract to 8 °Bx (55 ± 5 g/L) with water via a Leica refractometer (Fisher Scientific, Pittsburg, PA, USA). We dispensed each 100 mL sorghum extract in a 250-mL Erlenmeyer flask and sterilized it at 115 °C for 30 min before used.

### 2.2. Lactic Acid Tolerance Tests

In order to gain the seed culture, the activated *P. kudriavzevii* C-16 and its type strain ATCC 24210 were respectively inoculated in 50 mL sorghum extract medium at 30 °C with shaking (200 r/min) for 24 h. The fermentation media were respectively supplemented with 20, 30 and 40 g/L lactic acid (AR grade) and then the pH of the media were adjusted to 3.5 (the pH of the *Baijiu* fermented grains [6]) with 5 M NaOH. Afterwards, the fermentation medium was inoculated with the seed culture (2 × 10^7^ cells/mL). The fermentation was carried out at 30 °C for 72 h, with shaking at 200 r/min. During the fermentation process, samples were taken every 12 h to quantify biomass and analyze the concentration of lactic acid and higher alcohols. Uninoculated sorghum extract medium was used as a negative control. Do three parallels for each experiment.

### 2.3. Biomass Detection

The optical density (OD) was detected at 600 nm using Cytation 3 Cell Imaging Multi-Mode Reader (Agilent, Santa Clara, CA, USA).

The fermentation broth was centrifuged for 5 min at 9000× *g*. Subsequently, the supernate was eliminated and yeast cells were dried to constant weight in a drying oven at 80 °C, and then the dry weight of cells (DCW) were calculated.

### 2.4. Analysis Methods

The fermentation broth was centrifuged at 4000× *g* for 10 min to gain the supernate. Then we used a 0.22 µm hydrophilic MCE syringe filter to filter the supernate. Lactic acid was separated by an Aminex HPX-87H Ion Exclusion column (300 mm × 7.8 mm, 3 µm) and detected by high-performance liquid chromatography (HPLC; Waters, Suzhou, China) with a Refractive Index (RI) Detector (2414), based on the method described previously [9]. We used gradient concentration of lactic acid standard (99.9%, HPLC grade) and its peak area to establish the quantitative standard curve that could calculate the concentration of lactic acid in fermentation samples.

The higher alcohols in the supernate were extracted by HS-SPME (Automatic headspace sampling system; GERSTEL Inc., Baltimore, MD, USA) and analyzed by GC-MS (6890N GC system and 5975 mass-selective detector; Agilent, Santa Clara, CA, USA) based on the method described previously [10]. The mass spectra of unknown flavour components were identified by comparing with those in the Wiley 275.L (Agilent) database.

### 2.5. RNA Extraction and Sequencing

When the fermentation (initial fermentation medium supplemented with 30 g/L lactic acid) was carried out for 12 h and 24 h, the samples were centrifuged at 8000× *g* for 5 min at 4 °C, and the supernate was removed to obtain the cell pellets. The cell pellets were milled into fine powders in a precooled mortar with liquid nitrogen. The total RNA was extracted with RNAiso Plus reagent (TaKaRa, Dalian, China) containing chloroform and isopropyl alcohol. Then the obtained RNA was washed with 75% ethanol and dissolved in 50 µL of RNase-free water. The quality of the extracted RNA was evaluated with an Agilent bioanalyzer. Finally, the RNA samples were sequenced on an Illumina HiSeq 2500 platform with 100-nucleotide (nt)-long paired-end reads at the Biomarker Technologies Co., Ltd. (Beijing, China).

Raw data were processed by removing the rRNA sequences and low-quality reads (Q < 0.02). Then we got clean reads. The clean reads were compared with the reference genome (*P. kudriavzevii* CBS573). Cufflinks (version 2.1.1) was then used to predict the new transcripts. After gene expression levels were analyzed. Finally, databases such as KEGG, GO were used to obtain genes that were significantly differently expressed between *P. kudriavzevii* C-16 and its type strain ATCC 24210 under 30 g/L lactic acid stress (initial pH adjustment was 3.5).

### 2.6. Statistical Analysis

We used Microsoft Office Excel 2016, OriginPro 2021, and Adobe Illustrator CC 2020 to do all statistical analysis and graphs. Introduce the concept of FPKM (fragments per kilobase of exon model per million mapped reads) to measure gene expression levels. FPKM ≥ 500 is regarded as high expression, 15 ≤ FPKM < 500 are regarded as medium expression, 1 ≤ FPKM < 15 are regarded as low expression, and 0 ≤ FPKM < 1 are regarded as no expression. In a 2-tailed, unpaired Student’s *t* test, we considered the differences were significant when *p* values were lower than 0.05.

## 3. Results

### 3.1. Comparison of Lactic Acid Tolerance between P. kudriavzevii C-16 and the Type Strain P. kudriavzevii ATCC 24210

The lactic acid tolerance of the two *P. kudriavzevii* strains (see Materials and Methods) was respectively tested by fermentation experiment using different concentrations of lactic acid stress in sorghum extract medium. The results showed that *P. kudriavzevii* C-16 grew better under any given lactic acid stress (Figure 1). Under 30 g/L lactic acid stress, the final OD_600_ of the strain C-16 was 3.49 ± 0.14, which was 1.20 times as much as that of the type strain ATCC 24210 (Appendix A). What’s more, the strain C-16 and the type strain ATCC 24210 had the largest difference in end biomass under 30 g/L lactic acid stress, followed by 40 g/L lactic acid stress (Appendix A). The results indicated that as the initial lactic acid concentration of the medium increased, the growth of the strain C-16 and its type strain ATCC 24210 was more and more strongly inhibited, the biomass at the end of fermentation gradually decreased, and the growth rate gradually decreased. Compared with the type strain ATCC 24210, the strain C-16 showed higher biomass yield and faster growth rate under the same stress of lactic acid, indicating that the strain C-16 had better lactic acid tolerance and was more suitable for the high lactic acid stress production process.

Then we explored the lactic acid consumption capacity of *P. kudriavzevii* C-16 and the type strain ATCC 24210 under different concentrations of lactic acid stress. The results showed that the strain C-16 could metabolize more lactic acid and consume it faster than the type strain ATCC 24210 at any given lactic acid stress (Figure 2 and Appendix A). At 30 g/L lactic acid stress, there was 26.20 ± 0.12 g/L lactic acid in the culture medium when the fermentation of the type strain ATCC 24210 was over, while the strain C-16 only retained 19.27 ± 0.85 g/L lactic acid. In other words, the fermentation of the type strain ATCC 24210 consumed 3.80 g/L lactic acid for 72 h, while the strain C-16 used 10.73 g/L lactic acid, which was 2.83 times as much as that of the type strain ATCC 24210 (Figure 2 and Appendix A). Under 40 g/L lactic acid stress, the strain C-16 had 27.31 ± 0.68 g/L lactic acid residue, that is 12.69 g/L lactic acid used. It was indicated that the strain C-16 utilized lactic acid most under 40 g/L lactic acid stress (Figure 2c). At 12 h of fermentation, the rate of lactic acid consumption was the fastest (Appendix A) and the two strains had the largest difference in lactic acid consumption rate at 30 g/L lactic acid stress (Appendix A). Besides, the lactic acid consumption rate was also higher at 24 h of fermentation (Appendix A). When the fermentation medium was supplemented with 30 g/L lactic acid, the lactic acid consumption rate of the type strain ATCC 24210 was 0.070 g/L/h. The lactic acid consumption rate of the strain C-16 was 0.18 g/L/h, which was 2.55 times as much as that of the type strain ATCC 24210 (Appendix A). When the initial concentration of lactic acid in the fermentation medium was 30 g/L, the difference in lactic acid consumption capacity between the strain C-16 and type strain ATCC 24210 was the largest (Appendix A).

### 3.2. Transcriptional Profile of P. kudriavzevii C-16 under Lactic Acid Stress

We further studied the transcriptional profile of *P. kudriavzevii* C-16 under lactic acid stress. Moreover, the difference between the strain C-16 and the type strain ATCC 24210 under 30 g/L lactic acid stress was the largest, so the condition of 30 g/L lactic acid stress was selected for the subsequent study on gene differences at the transcription level. Then we compared the transcriptomes of the two *P. kudriavzevii* strains grown in the presence of lactic acid (30 g/L), at time points (12 h and 24 h) specifically chosen to explore the physiological mechanism of the strain C-16 in response to lactic acid stress. After controlling the quality of raw RNA-seq reads, 48.27, 56.87, 49.86, and 46.29 million clean reads were respectively obtained (samples of the strain C-16 cultured for 12 h and 24 h, and the type strain ATCC 24210 cultured for 12 h and 24 h) (Appendix A). Among them, the ratios of uniquely mapped reads to clean reads respectively were 95.24%, 94.93%, 94.64%, and 94.90%, with an average rate of 94.93% (Appendix A). Then, we detected about 5324 genes in all four samples. The number of genes with high expression, medium expression, low expression and no expression respectively was shown in Appendix A. Among them, the percent of genes above medium expression levels (FPKM ≥ 15) was respectively 79.09%, 77.31%, 80.97%, and 80.15%, with an average rate of 79.38%. Compared with the type strain ATCC 24210, 144 genes were identified as differentially expressed genes (DEGs) in *P. kudriavzevii* C-16 cultured for 12 h, of which 92 genes were significantly upregulated and 52 genes were significantly downregulated (Figure 3). In *P. kudriavzevii* C-16 cultured for 24 h, 154 genes were identified as DEGs, of which 96 genes were significantly upregulated and 58 genes were significantly downregulated (Figure 3).

When the strain C-16 was cultured under 30 g/L lactic acid stress for 12 h and 24 h, the scatter plot of KEGG enrichment of DEGs compared with the type strain ATCC 24210 was shown in Figure 4. The results of the analysis of KEGG enrichment showed that the DEGs were mainly concentrated on some pathways, such as biosynthesis of amino acids, ABC transporters, pyruvate metabolism, biosynthesis of secondary metabolites, 2-oxocarboxylic acid metabolism, MAPK signalling pathway-yeast, and glycerolipid metabolism.

### 3.3. Response Mechanisms to Lactic Acid Stress in P. kudriavzevii C-16

We further analyzed the DEGs in several metabolic pathways mainly enriched in the KEGG classification. As shown in Figure 5, further analysis of DEGs suggested that *P. kudriavzevii* degraded lactic acid through the pyruvate metabolic pathway. It was also mentioned in the previous article [9]. Under lactic acid stress, the genes involved in the degradation of lactic acid in the strain C-16 were significantly upregulated compared with the type strain ATCC 24210 (Appendix A). Among them, *lldD* encodes L-lactate dehydrogenase (EC:1.1.2.3), which can convert L-lactate to pyruvate. Compared with the type strain ATCC 24210, the *lldD* of the strain C-16 was upregulated by 6.13 times as culturing for 12 h, and 11.02 times as culturing for 24 h. *MAE1* encodes malate dehydrogenase (EC:1.1.1.38) and *PCK1* encodes phosphoenolpyruvate carboxykinase (EC: 4.1.1.49). Compared with the type strain ATCC 24210, *MAE1* and *PCK1* of *P. kudriavzevii* C-16 were respectively upregulated by 2.33 and 2.64 times at 12 h of fermentation. Therefore, the pathway, in which pyruvate produced (S)-malate, then (S)-malate was further converted to phosphoenolpyruvate with oxaloacetate, and finally to generate pyruvate, was strengthened. *LEU4* encodes 2-isopropylmalate synthase (EC: 2.3.3.13), which catalyzes the formation of acetyl-CoA into 3-carboxy-3-hydroxy-4-methylpentanoate. Compared with the type strain ATCC 24210, *LEU4* of the strain C-16 was upregulated by 2.60 times at 12 h and 3.01 times at 24 h. Under lactic acid stress, compared with the type strain ATCC 24210, the strain C-16 degraded lactic acid more efficiently, and the expression of lactate dehydrogenase at 24 h was higher than that at 12 h. After lactic acid was converted to pyruvate, one part produced 3-carboxy-3-hydroxy-4-methylpentanoate, which was involved in the biosynthesis of leucine.

Besides, transcriptome analysis of *P. kudriavzevii* C-16 found that in addition to the gene of lactate dehydrogenase involved in lactic acid degradation, under the stress of 30 g/L lactic acid, there was one gene *SNQ2* encoding membrane transporter protein significantly upregulated compared with the type strain ATCC 24210 (Figure 5). Under 30 g/L lactic acid stress, three transcripts of *SNQ2* of *P. kudriavzevii* C-16 were all significantly upregulated compared with the type strain ATCC 24210 (Appendix A). Among them, C5L36_0C11740 was upregulated by 31.13 times at 12 h and 54.80 times at 24 h. The up-regulation of this transcript gene is more than 10 times that of the other two transcripts.

Glutamate metabolic pathway plays an important role in maintaining the cell viability of microorganisms under acid stress [11]. In the pathway, we found that gene *GAD1* and gene *UGA2* were upregulated. *GAD1* encodes glutamate decarboxylase (GAD) (EC: 4.1.1.15) that can convert L-glutamate and protons to γ-aminobutanoate (GABA) and carbon dioxide. Under 30 g/L lactic acid stress, *GAD1* of the strain C-16 was upregulated by 2.45 times when cultured for 12 h and 1.74 times when cultured for 24 h (Appendix A). As shown in Figure 5, when the cell was at a low pH, GAD catalyzes the conversion of L-glutamate to GABA to consume intracellular protons. Reduce the concentration of protons in the cell, thereby protecting the cell from acid damage [12]. Additionally, GABA is less acidic than glutamate, which can alkalize the environment. GABA can be further converted into succinate semialdehyde, and then it is converted into succinic acid by the catalysis of succinate semialdehyde dehydrogenase. Succinic acid, the final product of GABA metabolism, can be replenished to the TCA cycle, which provides the required carbon skeleton for the cycle. *UGA2* encodes succinate semialdehyde dehydrogenase (EC: 1.2.1.16). Under 30 g/L lactic acid stress, *UGA2* of the strain C-16 was upregulated by 2.24 times when cultured for 12 h and 1.75 times when cultured for 24 h (Appendix A).

Furthermore, it has been found that lactic acid induced the response of the genes of the amino acid synthesis pathway. Compared with the type strain ATCC 24210, *P. kudriavzevii* C-16 enhanced the biosynthesis of leucine, valine, and isoleucine in a fermentation environment with a high concentration of lactic acid stress (Figure 5). During the biosynthesis of leucine, the expression of four genes was upregulated. Among them, *ILV5* encodes ketol acid reductoisomerase (EC: 1.1.1.86), *LEU4* encodes 2-isopropylmalate synthase (EC: 2.3.3.13), *LEU1* encodes 3-isopropylmalate dehydrogenase (EC: 4.2.1.33), and *LEU2* encodes 3-isopropylmalate dehydrogenase (EC: 1.1.1.85). During the biosynthesis of valine, the expression of the *ILV5* gene was upregulated. During the biosynthesis of isoleucine, the expression of *LEU2* and *ILV5* was upregulated. Under lactic acid stress, the gene *ILV5* of the strain C-16 was respectively upregulated by 2.66 and 1.97 times at 12 h and 24 h, and *LEU4* was respectively upregulated by 2.60 and 3.01 times at 12 h and 24 h. *LEU1* was respectively upregulated by 4.49 and 4.43 times at 12 h and 24 h, and the *LEU2* gene was upregulated by 2.09 and 1.81 times at 12 h and 24 h, respectively (Appendix A).

### 3.4. Higher Alcohols Production

Amino acids are the precursors of higher alcohols and play an important role in the fermentation process of Chinese *Baijiu* [13]. Proper concentration of higher alcohols can make the *Baijiu* taste harmonious and increase the sweetness and aftertaste of the *Baijiu* [10]. Therefore, we further compared the higher alcohols production of the strain C-16 and its type strain ATCC 24210 under 30 g/L lactic acid stress (Figure 6). At the end of the fermentation, three kinds of higher alcohols, isobutyl alcohol, isoamyl alcohol, and phenethyl alcohol, were detected. Under lactic acid stress, the type strain ATCC 24210 produced 418.97 ± 5.02 μg/g DCW (dry cell weight) total higher alcohols. Compared with the type strain ATCC 24210, the production of higher alcohols of *P. kudriavzevii* C-16 was 448.10 ± 37.90 μg/g DCW, which was 1.07 times as much as that of the type strain ATCC 24210. The yield of isobutyl alcohol of the type strain ATCC 24210 was 6.66 ± 0.34 μg/g DCW. Compared with that, the isobutyl alcohol yield of the strain C-16 was significantly increased (*p* < 0.05), which could produce 14.84 ± 1.95 μg/g DCW and was 2.23 times as much as that of the type strain ATCC 24210. The type strain ATCC 24210 produced 177.97 ± 4.40 μg/g DCW isoamyl alcohol. Compared with that, the yield of isoamyl alcohol of the strain C-16 was significantly improved (*p* ˂ 0.01), which could produce 216.55 ± 7.97 μg/g DCW and was 1.22 times as much as that of the type strain ATCC 24210. *P. kudriavzevii* C-16 and the type strain ATCC 24210 respectively produced 216.71 ± 32.36 μg/g DCW and 234.34 ± 0.52 μg/g DCW phenethyl alcohol, and there was no significant difference in the yield of phenethyl alcohol between the two strains (*p* > 0.05).

## 4. Discussion

*P. kudriavzevii* C-16 was more suitable for the Chinese *Baijiu* brewing system because of its adaptive evolution. By comparing the lactic acid tolerance of the strain C-16 and its type strain ATCC 24210, the OD_600_ of the strain C-16 was respectively 1.14, 1.20, and 1.16 times more than that of the type strain ATCC 24210 under 20, 30, and 40 g/L lactic acid stress at the end of fermentation. Moreover, the results showed that the lactic acid consumption and consumption rate of the strain C-16 were also higher than that of the type strain ATCC 24210. Under 20, 30, and 40 g/L lactic acid stress, the degradation of lactic acid of the strain C-16 was 1.37, 2.83, and 2.04 times more than that of the type strain ATCC 24210, respectively. Similar domestication phenomenon also appeared in other food or beverage fermentation environments. For example, the use for millennia of sulfite as a preservative in wine production is conducive to the production of corresponding resistant mutation in wine yeast strains, but it has not been detected in wild strains [14,15,16]. *Planococcus maritimus* XJ11, isolated from traditional fermented shrimp paste, exhibits great adaption to low temperature, low salinity, and alkaline environment [17]. Various independent non-sense mutations of genes, related to the production of 4-vinyl guaiacol (4VG), have been found in many industrial (and especially beer) yeasts, but they have not been found in biofuels or non-industrial isolated yeasts [1,2]. It is indicated that a specific fermentation environment can lead to the adaptive evolution of strains that have been in it for a long time, and the mutant phenotype is almost undetected in the natural environment.

The root of the phenotypic differences in the adaptive evolution of strains is genetic differences [18]. Based on transcriptomics analysis, the effect of a high concentration of lactic acid stress on the strain C-16 was explained at the gene transcription level. Compared with the type strain ATCC 24210, it enhanced the degradation of lactic acid through the pyruvate metabolic pathway in the strain C-16 under the high concentration of lactic acid stress. Among them, the *lldD* (encodes L-lactate dehydrogenase) of the strain C-16 was significantly upregulated by 6.13 times as culturing for 12 h, and 11.02 times as culturing for 24 h. In other microorganisms, the L-lactate dehydrogenase encoded by *lldD* also plays a vital role in the degradation of L-lactate. For example, in *Corynebacterium glutamicum*, *lldD* was confirmed to encode the only L-lactate dehydrogenase, which was essential for the growth of *C. glutamicum* on L-lactate [19]. The mutant that knocked out the *lldD* gene could not grow with L-lactate as the sole carbon source, providing evidence for the indispensable role of L-iLDH in L-lactate utilization in *Pseudomonas stutzeri* SDM [20]. BAKERS’ yeast L-lactate dehydrogenase or cytochrome *b_2_* (EC 1.1.2.3) catalyzes the oxidation of L-(+)-lactate to pyruvate [21].

The expression of ABC transporters was also enhanced, which improved the efficiency of cell efflux system. Under the stress of 30 g/L lactic acid, C5L36_0C11740, one transcript of *SNQ2* (encoding membrane transporter protein), was significantly upregulated by 31.13 times at 12 h and 54.80 times at 24 h compared with the type strain ATCC 24210 (Figure 5). It has been found that the cell efflux system has been proved to be an important mechanism for cells to tolerate extracellular toxic substances and harmful intracellular metabolites [22]. In *S. cerevisiae*, there are a large number of ABC transporters involved in the stress response of carboxylic acids [23]. The transporter protein encoded by gene *SNQ2* belongs to the ABC transporter protein family [24,25]. In *S. cerevisiae*, the protein encoded by *SNQ2* is similar to the protein encoded by the multi-drug resistance gene *YDR1*, and they both belong to the ABC transporter protein family [26].

Furthermore, the glutamate metabolic pathway was enhanced, which improved the efficiency of H^+^ consumption. The glutamate metabolic pathway has been confirmed to be acid-resistant in several bacterial genera. For example, it has been described as the most important acid-resistant mechanism in the highly acid-resistant bacteria *Escherichia coli* [27,28]. In plants, the glutamate metabolic pathway is expressed in response to various stress conditions (such as temperature, hypoxia, or elevated Ca^2+^ levels) [29].

Moreover, the results showed that during the synthesis of leucine and valine, the transcription of four enzymes was significantly up-regulated, indicating that the biosynthesis efficiency of leucine and valine in *P. kudriavzevii* C-16 is higher than that in the type strain ATCC 24210 under lactic acid stress. Leucine and valine are the precursors of important flavour substances, isoamyl alcohol, and isobutyl alcohol, respectively. The production of isoamyl alcohol and isobutyl alcohol in *P. kudriavzevii* C-16 under lactic acid stress is 1.43 and 2.81 times more than that in the type strain ATCC 24210, respectively. Therefore, the results of higher alcohols analysis were consistent with the upregulation of gene transcription. In addition, amino acids are also an important resource of lactic acid bacteria (LAB), and their utilization has many physiological functions, such as intracellular pH control, production of metabolic energy or redox capacity, and resistance to stress [30].

## 5. Conclusions

In conclusion, this work showed the response mechanism to lactic acid stress in *P. kudriavzevii* C-16 with the type strain ATCC 24210 as a control. Our study suggested that the synergistic effect of pyruvate metabolic pathway, ABC transporter proteins, glutamate metabolic pathway, and the biosynthetic pathway of leucine and valine can resist to high concentrations of lactic acid in the strain C-16. Further work should focus on investigating the specific functions of the genes involved these pathways. Thus, it can be used to guide the effective utilization of *P. kudriavzevii* in food and beverage fermentation.

## Figures and Tables

**Figure 1 foods-11-02725-f001:**
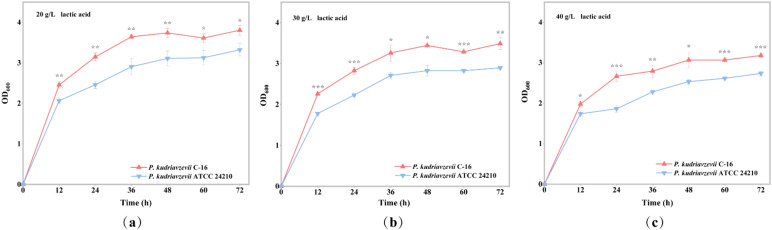
Biomass of *P. kudriavzevii* C-16 and the type strain ATCC 24210 under different concentrations of lactic acid. (**a**) Under 20 g/L lactic acid stress; (**b**) under 30 g/L lactic acid stress; (**c**) under 40 g/L lactic acid stress. The initial pH of the culture medium was adjusted to 3.5. * *p* < 0.05, ** *p* < 0.01, *** *p* < 0.001.

**Figure 2 foods-11-02725-f002:**
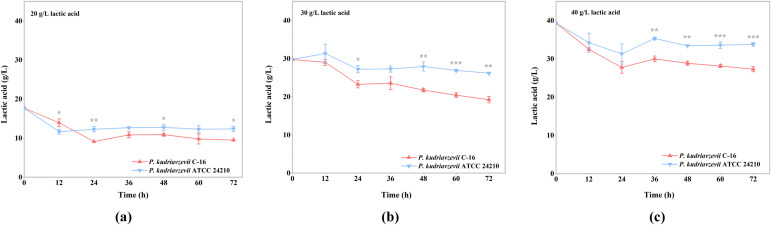
Lactic acid degradation performance of *P. kudriavzevii* C-16 and the type strain ATCC 24210 under different concentrations of lactic acid stress. (**a**) Under 20 g/L lactic acid stress; (**b**) under 30 g/L lactic acid stress; (**c**) under 40 g/L lactic acid stress. The initial pH of the culture medium was adjusted to 3.5. * *p* < 0.05, ** *p* < 0.01, *** *p* < 0.001.

**Figure 3 foods-11-02725-f003:**
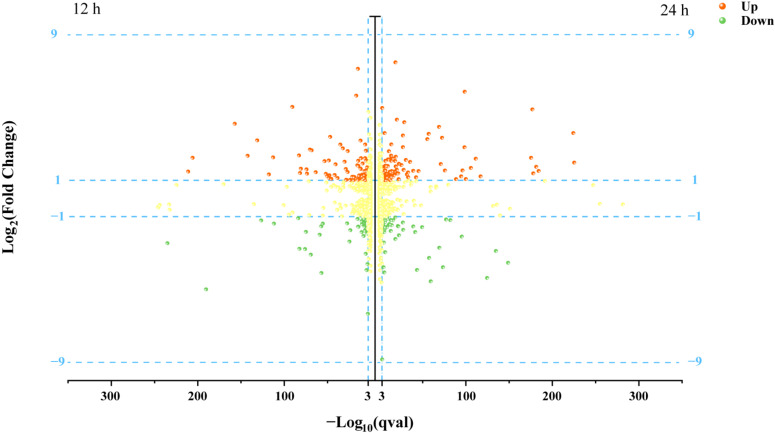
Differential expression genes of *P. kudriavzevii* C-16 and the type strain ATCC 24210 under 30 g/L lactic acid stress. Imaginary lines indicate inclusion criteria for log_2_(fold change) ≥ 1 or ≤−1 and −log_10_(qval) > 3. Red, up-regulated expressed genes in *P. kudriavzevii* C-16; green, down-regulated expressed genes; yellow, genes that did not meet the inclusion criteria.

**Figure 4 foods-11-02725-f004:**
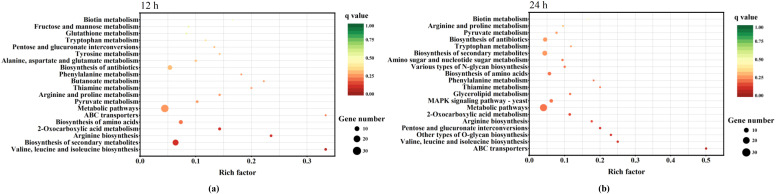
KEGG pathway enrichment scatterplot of differentially expressed genes of *P. kudriavzevii* C-16 and the type strain ATCC 24210 under lactic acid stress. (**a**) Cultured for 12 h; (**b**) Cultured for 24 h.

**Figure 5 foods-11-02725-f005:**
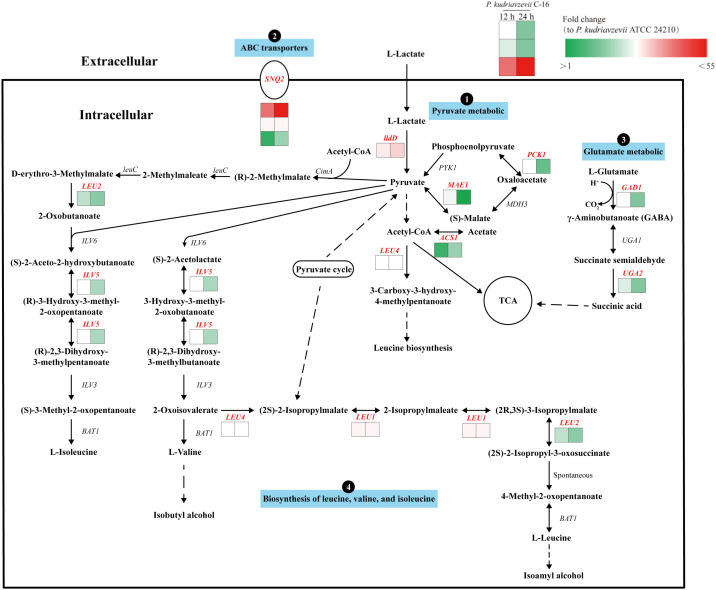
Pathways of *P. kudriavzevii* C-16 response to lactic acid stress. Red letters mean upregulated genes. Transcripts of *P. kudriavzevii* C-16 are shown near the pathway as a heat map, based on fold-changes in transcript levels relative to the type strain ATCC 24210. Colour legend is shown at the top right of the map.

**Figure 6 foods-11-02725-f006:**
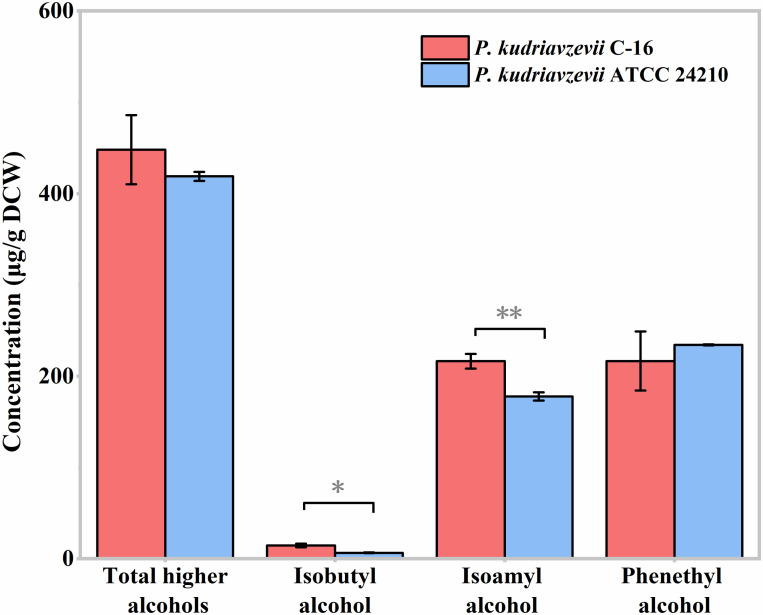
Higher alcohols production of *P. kudriavzevii* C-16 and its type strain ATCC 24210. Under 30 g/L lactic acid stress and the initial pH of the culture medium was adjusted to 3.5. “Total higher alcohols” means the sum of isobutyl alcohol, isoamyl alcohol, and phenethyl alcohol. * *p* < 0.05, ** *p* < 0.01.

## Data Availability

The data presented in this study are available on request from the corresponding author.

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
