# Peer review of "Transcriptional Profiling Reveals Adaptive Response and Tolerance to Lactic Acid Stress in Pichia kudriavzevii"

_foods, 2022, doi:10.3390/foods11182725_

Round 1

Reviewer 1 Report

I have revised: Transcriptional profiling reveals adaptive response and toler-ance to lactic acid stress in Pichia kudriavzevii

The work is very good, the results, discussions and conclusions are very complete, however it contains some omissions in the method that should be clarified, it also contains too many self-citations. 

Line 85: What did you mean with “seed culture”? Please Clarify in the text.

Line 85-87: It is not described how the inoculum was prepared or how much of the organism was deposited. Please clarify in the text.

Line 69-71: Is not clear when the material where incubated at 30oC, or Why? For further analyses? For fresh cultures? Clarify un text.

Line 95-98: It is mentioned twice that the broth was centrifuged. Are they separate processes? Please separate each analysis process by subtopics.

Line 112: What do you mean by "The cell pellets" with what process were they obtained? clarify in text.

Please check the self citations there are a lot, please remove it and add others than support your work.

Reviewer 2 Report

The authors presented preliminary results regarding the use of Pichia kudriavzevii as a starter culture for the Chinese Baijiu brewing system

L48 The production of aflatoxin by A. flavus is conditionate to intrinsic and extrinsic conditions. The authors should describe what conditions have no observed aflatoxin production.

L86-89 I did not understand well. The authors first adjusted the pH of the medium to 3.5 and then added lactic acid? or partially neutralized the medium?

This section is very confusing. What was the initial pH of the sorghum extract medium before adjusting? Was it less than 3.5? How pure was the lactic acid used? New medium was prepared for each replicate?

L95 Replace ‘rpm’ with ‘g’

L140 In which section was the method for microbial counting described

Figure 1 The absorbance values are too high for microbial suspension, usually no more than score 2 is achieved.

Figure 2 - What about the concentration of lactic acid in the controls?

Round 2

Reviewer 1 Report

The authors addressed all comments correctly. 

Reviewer 2 Report

Accept